# Piloting a Measure of Segregation at the Census Tract Level: Associations with Place and Racial/Ethnic Disparities in Life Expectancy

**DOI:** 10.3390/ijerph21050613

**Published:** 2024-05-11

**Authors:** Katherine Kitchens, Genevieve Graaf

**Affiliations:** School of Social Work, University of Texas at Arlington, 501 W. Mitchell Street, Arlington, TX 76019, USA; genevieve.graaf@uta.edu

**Keywords:** place, segregation, segregation measure, health disparities, racial/ethnic health disparities

## Abstract

This study considers residential segregation as a critical driver of racial/ethnic health disparities and introduces a proxy measure of segregation that estimates the degree of segregation at the census tract level with a metric capturing the overrepresentation of a racialized/ethnic group in a census tract in relation to that group’s representation at the city level. Using Dallas, Texas as a pilot city, the measure is used to investigate mean life expectancy at birth for relatively overrepresented Hispanic, non-Hispanic white, non-Hispanic Black, and Asian census tracts and examine for significant differences between mean life expectancy in relatively overrepresented census tracts and that group’s mean life expectancy at the state level. Multivariable linear regression analysis was utilized to assess how segregation measured at the census tract level associates with life expectancy across different racialized/ethnic groups, controlling for socioeconomic disparities. This study aimed to expose the need to consider the possibility of neighborhood mechanisms beyond socioeconomic characteristics as an important determinant of health and draw attention to the importance of critically engaging the experience of place in examinations of racial and ethnic health disparities. Multivariable linear regression modeling resulted in significant findings for non-Hispanic Black, non-Hispanic white, and Asian groups, indicating increased census tract-level life expectancy for Black and white residents in highly segregated census tracts and decreased life expectancy for residents of tracts in which the Asian community is overrepresented when compared to state means. Unadjusted models demonstrated socioeconomic inequities between first and fourth quartile census tracts and pointed to the importance of mixed methods in health disparities research and the importance of including the voice of community members to account for places of daily lived experience and people’s relationships with them.

## 1. Piloting a Measure of Segregation at the Census Tract Level: Associations with Place and Racial/Ethnic Health Disparities in Life Expectancy

The United States has a long history of discriminatory housing practices that have led to persistent patterns of racialized/ethnic residential segregation, which are associated with structural inequities that have consequences spanning multiple generations [1,2,3]. In metropolitan areas, spatial patterns continue to adhere to historic lending boundaries established by the discriminatory redline maps that were sanctioned by the Federal Housing Administration in the 1930s [4,5]. Current research has shown a statistically significant correlation between areas that were formerly redlined and various neighborhood characteristics, such as the increased presence of minoritized communities, poverty, social vulnerability, poor mental health, and risk of morbidity in COVID-19 patients [5,6]. Consequently, the historical practice of redlining exemplifies the place-bound nature of racialized divisions [7] and the enduring impact of structural racism on the health and well-being of minoritized communities.

Segregation has been recognized as a root cause of racial/ethnic health disparities due to its role in shaping multiple factors crucial for health promotion and disease prevention [8,9]. Although the detrimental relationship between segregation and health continues to be identified as a primary driver of Black–white health disparities [10,11,12], only a limited number of studies have investigated variations in segregation’s relationship with health outcomes among different minoritized communities [13]. Moreover, existing studies often rely on formal indices that measure segregation across large geographical areas, resulting in a critical gap in examining segregation at the neighborhood level [13,14]. This reliance on large-scale measures impedes the study of the impacts of segregation as a daily lived experience.

This study introduces a proxy measure of segregation that estimates the degree of segregation at the neighborhood level using a metric that captures the overrepresentation of a racialized/ethnic community in a census tract compared to that group’s representation at the city level. By measuring segregation at a more granular level (i.e., census tract), this approach provides an opportunity to capture neighborhood context while considering a group’s distribution within the city’s total population. Exposing disparities in life expectancy among racialized/ethnic groups by examining patterns of over- and underrepresentation of group members in residential tracts has the potential to offer an alternative perspective in examining associations between structural mechanisms within the lived environment and health. Using Dallas, Texas as a pilot city, this measure is employed to investigate the association between the overrepresentation of Hispanic, non-Hispanic white, non-Hispanic Black, and Asian communities at the neighborhood level and neighborhood rates of life expectancy at birth, considering how they differ from each group’s life expectancy at the state level. The aim of this study is to highlight the need to consider neighborhood mechanisms beyond socioeconomic characteristics as key determinants of health and to draw attention to the importance of engaging the experience of neighborhoods when examining racial and ethnic health disparities.

## 2. Background

The practice of redlining originated with the establishment of the Home Owners’ Loan Corporation (HOLC) in 1933. This government-sponsored corporation, created during the New Deal era, was initially intended to prevent home foreclosures by refinancing defaulted mortgages. However, the HOLC operated under the assumption that the presence of homeowners who identified as Black, regardless of economic class, diminished the value of white-owned property [4]. Consequently, the HOLC created color-coded maps of every major metropolitan area in the United States, denying federally insured home mortgages to buyers in red-coded areas (i.e., predominantly Black neighborhoods) while supporting buyers’ mortgages in green-coded areas (i.e., predominantly white neighborhoods). This federal policy sought to disinvest in communities deemed hazardous or predicted to be deteriorating based on racial composition while subsidizing suburban growth through federally insured home mortgages. This history of de jure segregation in the United States exemplifies institutionalized injustice that perpetuates place-based divisions among racialized groups through the inheritance of policies that “racialize space and spatialize race” [7].

The withdrawal of lending sources in urban areas inevitably resulted in neighborhood deterioration [15]. It established persistent patterns of segregation, which has been recognized as a critical driver of racial inequities operating through multiple pathways related to public and private disinvestment. Segregation affects communities of color as a determinant of (1) socioeconomic status and the probability of living in areas of concentrated poverty, (2) lack of access to quality health care, education, employment opportunities, and housing stock, (3) exposure to high levels of neighborhood violence and crime, and (4) exposure to psychosocial stressors and environmental hazards (Williams et al., 2019). Consequently, segregation implicates racism and oppression as spatial acts, illustrating how places are socially and differentially constructed and geographies of subjugation and displacement are perpetuated [16]. These *power geometries* [17] delineate inequities associated with place, revealing how the inequitable distribution of power and resources is interwoven as lines of color into the fabric of American life [18,19].

## 3. Health Disparities Research

The disproportionate impact of the SARS-CoV-2 pandemic on communities of color has sparked considerable interest in the study of health disparities, but the root cause of these disparities has been a longstanding question for public health and social work researchers. However, the literature primarily focuses on individual-level factors (e.g., socioeconomic status, genetics, and health behaviors) as fundamental causes of health disparities [9]. For instance, a 2019 study by Alvidrez and colleagues [20] discovered that place-based factors related to the built environment and neighborhood, or community-level factors, are underrepresented in health disparities research funded by National Institute on Minority Health and Health Disparities (NIMHD) R01 grants. The predominant emphasis on individual-level factors fails to consider that deep-seated structural disparities—such as those revealed by the pandemic—are place-based and have place-based impacts, affecting people of color within a unique lived environment.

Over the past 15 years, the shift in focus to segregation as a key determinant of health has been a significant area of interest in health disparities research. These studies have enhanced the understanding of the impact of distal and intermediate influences on the health of minoritized communities. However, because the predominant measures used in segregation-focused research are indices that measure segregation across large geographical areas [21], only a limited number of studies have investigated the association between segregation at the neighborhood level and the lived experience of place [14]. The structural inequities of the lived environment implicated as the mediating factor in segregation’s adverse relationship with health—concentrated poverty, lack of access to critical resources, and exposure to psychosocial stressors and environmental hazards [2]—are concealed by such large-scale measures. Combined with a lack of research exploring health disparities outside the Black–white binary [13], current methods obscure the examination of the lived experience of place and segregation’s differential association with the health of diverse communities.

## 4. Residential Segregation and Health

Health disparities are the metric used to gauge progress toward health equity among minoritized communities. These disparities are (1) systematic and preventable, (2) originate from discrimination and marginalization, and (3) perpetuate social disadvantage and vulnerability [22]. The U.S. Department of Health and Human Services Office of Minority Health (2020) reports that the national life expectancy at birth for individuals identifying as non-Hispanic Black is 77.0 years, which is lower than the 80.6 years, 80.7 years, and 82.1 years for those identifying as non-Hispanic white, Asian, and Hispanic, respectively [23]. The OMH also estimates that 21.2% of people who identify as Black and 17.2% of people who identify as Hispanic live at the federal poverty level, compared to 9.0% and 9.6% of those who identify as white or Asian, respectively. However, these economic disparities alone do not fully explain the discrepancy in life expectancy between Black and Hispanic community members.

Although structural inequities related to social stratification are frequently identified as the mechanisms through which segregation affects health, research suggests that some minoritized groups experience better health outcomes despite neighborhood context and socioeconomic status. The Hispanic Epidemiological Paradox posits that Americans who identify as Hispanic tend to have better health and longer lifespans compared to those who identify as white, despite lower socioeconomic status and obstacles to accessing quality education and healthcare [24,25]. A 2015 report by the U.S. Centers for Disease Control and Prevention (CDC) [26] revealed that the Hispanic population in the U.S. had a 24% lower all-cause mortality risk compared to the white population, even though 40% of respondents reported lacking health insurance coverage. Conversely, when examining Black–white disparities in hypertension rates (40.3% and 27.8%, respectively), it is noteworthy that Black populations in Caribbean countries and Africa have lower hypertension rates than people who identify as white in the United States [27]. These findings underscore the importance of shifting the study of health disparities away from individual-level characteristics and towards social systems and structures, including place and people’s relationship with it as determinants of health.

While it is essential to acknowledge that segregation is a determinant of the likelihood of living in concentrated poverty and lacking access to resources and opportunities [8,9,28], the variability in health outcomes among different groups necessitates an approach that views segregation as an experience of discrimination. The manifestation of power and racialization through segregation—along with the place-related identity development and chronic stress that may ensue—demands the development of methods that differentiate between experiences of segregation within and between communities. For instance, some researchers hypothesize that ethnic enclaves may offer health-protective benefits in the form of social capital, such as strong community ties and social cohesion [9,13,29]. Although intragroup differences among Hispanic communities have been identified, research suggests that health-protective factors for Mexican Americans residing in densely populated Mexican American neighborhoods counterbalance elevated poverty levels and other structural inequities [29]. Consequently, the primary focus on the Black–white binary, which largely overlooks comparisons with Hispanic and Asian communities [13], fails to recognize the significance of examining the experience of place as a site of power and identity-making and how that lived experience influences the health of different communities.

## 5. Measuring Segregation

The existing empirical research linking segregation to health disparities heavily relies on well-established standardized measures of segregation. Massey and Denton [21], in their influential article, outline five dimensions along which segregation can be measured: (1) evenness assesses the extent to which racialized communities are unevenly distributed across areal units, (2) exposure gauges the potential for contact between members of different racialized communities, (3) concentration evaluates the relative geographical space occupied by a racialized community within an urban area, (4) centralization determines the degree to which a racialized group is spatially situated near the central business district of a city, and (5) clustering measures the extent to which clusters of racialized communities are spatially adjacent to one another.

Considering the field of segregation research to be in a state of theoretical and methodological disorder, Massey and Denton evaluated twenty segregation indices based on the five axes described above. They used factor analysis to determine the single best indicator for each dimension of segregation. These five indices have served as the standard segregation measures for over thirty years [30,31]. Among them are the Index of Dissimilarity measuring evenness, the *P*^*^ Indices (i.e., Interaction or Isolation) measuring exposure, the Relative Concentration Index measuring concentration, the Absolute Centralization Index measuring centralization, and the Index of Spatial Proximity measuring clustering.

According to Yang et al. [13], segregation-focused health disparities research predominantly employs measures of evenness and exposure. The Index of Dissimilarity, a measure of evenness and a cornerstone of segregation research, represents the proportion of racialized community members who would need to relocate to a different census tract to achieve an even distribution of community members within a large geographical area (e.g., county, metropolitan statistical area (MSA)). On the other hand, exposure indices indicate the likelihood of contact between racially different community members. The Interaction Index, one of the most widely used exposure indices, directly measures the potential for contact between majority and minoritized communities. Another frequently employed exposure index is the Isolation Index, which indicates the percentage of same-group population in the census tract where the average community member resides.

While the Index of Dissimilarity, the Interaction Index, and the Isolation Index are well-established and widely used measures of segregation across large geographical areas, there remains a significant gap in understanding the impact of segregation on the physical environment at the neighborhood level. This gap persists because most of the published literature calculates segregation by describing the distribution of communities across micro-units (i.e., census tracts) within a larger macro-area (e.g., county, MSA) [9]. Examining health outcomes using such broad measures statistically obscures subunit conditions [32] and overlooks neighborhood characteristics that contribute to the relationship between segregation and health disparities—such as concentrated poverty, lack of access to essential resources, and exposure to psychosocial stressors and environmental hazards [2]. For instance, according to the most recent American Community Survey (ACS) 5-year estimates, individuals who identify as Asian constitute 3.75% of the total population of Dallas, Texas [33]. Using the overall high school completion rate for the Asian community in Dallas (85.56%) as an indicator of educational equity conceals place-based disparities, such as those found in a densely populated census tract where the Asian population accounts for 21.55% of the total population, and the high school completion rate for the Asian community is only 4.70%.

Measuring segregation at the census tract level enables analyses to capture and account for the social and structural determinants of health that stem from segregation and other forms of institutionalized discrimination. However, Kramer and Hogue point out that the few studies that used the census tract as a proxy for neighborhood treated the racial/ethnic composition of a tract as an isolated geometry [14]. This approach fails to provide an understanding of how groups are distributed in relation to the larger city or MSA, and it also lacks a reference point against which to compare the racial composition of the census tract.

To address the limitation in the existing understanding of segregation and health outcomes, this study considered Kramer and Hogue’s critique when developing a segregation measure that uses the residential tract as the primary unit of analysis. The study acknowledged that a small-scale measure is essential for examining neighborhood context; however, it also recognized that to be an effective measure, it must consider a community’s distribution within the total population. By focusing on the mechanisms identified as the link between segregation and individual health and examining the resulting socioeconomic disparities at a more granular level, the current study observed differential relationships between communities and the places where they live.

## 6. Methods

The primary objective of this study was to develop and pilot a proxy measure that estimates the degrees of segregation at the census tract level. This measure was then employed to identify census tracts in which groups were most overrepresented and assessed for significant differences in the mean life expectancy of a group within the most overrepresented census tracts compared to that group’s mean life expectancy at the state level, while accounting for socioeconomic disparities. The study aimed to answer two research questions: (1) Is there a statistically significant difference between the mean life expectancy at birth of census tracts where a community is relatively overrepresented and that community’s mean life expectancy at the state level? (2) Do these differences vary across racialized and ethnic communities?

To account for the unique political, social, and economic structures in the state of Texas that may impact the health and well-being of all Texans, census tract estimates were compared with state estimates rather than national estimates. The study hypothesized that increases in the relative overrepresentation of a given racialized/ethnic community in a census tract would be associated with significant differences in the tract’s mean life expectancy when compared to that community’s life expectancy at the state level. Furthermore, it was hypothesized that the nature and magnitude of this difference would vary across minoritized communities.

## 7. Study Setting

To examine the association between the overrepresentation of a group in a residential tract and mean life expectancy at birth in the United States, this study initially aimed for a broad conceptualization. However, the extensive geographical range of such an undertaking was beyond the scope of the study. Consequently, it was necessary to limit the scope by piloting the measure in a single U.S. city. The study setting encompasses all individuals reported to reside in census tracts located in Dallas, Texas. The city includes a total of 303 individual census tracts (*n* = 303).

Dallas was chosen as the study setting for several reasons, presented here in no particular order. First, the research team has familiarity with Dallas, including knowledge of its history and an understanding of the nuances of neighborhood composition throughout the city. Second, Dallas County ranks 26th out of the 52 most segregated urban counties in the United States [34]. (See Table 1 for the values of formal segregation indices for Dallas, Texas). Furthermore, according to the CDC’s Social Vulnerability Index scores, the boundaries of the most vulnerable Dallas census tracts align with the historic lending patterns established by the 1930s redline maps [35].

## 8. Key Variables

### 8.1. Predictor Variable: The Representation Index

The predictor variable in this study was the degree of segregation at the census tract level. Unlike standard indices of segregation, this measure considers the census tract as the macro unit of analysis, determining a group’s segregation degree through a metric that compares a community’s relative representation in a census tract to its overall representation in the city where it is nested. According to the U.S. Census [36], a census tract is a small, relatively stable statistical county subdivision, ideally populated by 4000 people. In this study, the census tract was used as a proxy for neighborhood. The maintenance of census tract boundaries over time allows for statistical comparisons from one census to the next, enabling the measure’s use for future longitudinal studies. Ethical considerations related to the use of census tract data are mitigated by the U.S. Census Bureau’s stringent privacy protocols, which anonymize and aggregate data to ensure individual privacy is upheld.

As shown in Table 2, population estimates from the 2015–2019 American Community Survey (ACS) were used to identify the percentage of a community residing in a census tract and city. The U.S. Census Bureau collected these population estimates between 1 January 2015 and 31 December 2019, and included data for all populations, disaggregated by racialized/ethnic community, regardless of geographic size. The decision to use the ACS five-year estimates was based on the guidance provided by the U.S. Census Bureau [37]. First, the collected data represent the most reliable estimates compared to the ACS one-year supplemental and three-year estimates. Second, the five-year estimates are the most suitable choice when precision is crucial in analyzing relatively small populations, such as census tracts. Finally, the 2015–2019 population estimates overlap with the collection dates of the life expectancy data used in this study.

The measure calculates the difference between the proportion of a census tract’s total population represented by a specific racialized/ethnic group (A) and the proportion of the city’s total population represented by that same racialized/ethnic group (B). This difference (C) provides an estimate of the community’s degree of segregation (i.e., relative representation) at the census tract level.
(1)C=A−B

Positive values suggest that a community is relatively overrepresented in a tract, while negative values indicate underrepresentation. Values close to zero imply that a community’s representation in the tract is relatively proportional to its representation in the city’s overall population. Table 2 provides the data sources used to calculate the degree of segregation.

This measure is scalable as it addresses a limitation of standard indices measuring segregation; it accounts for the heterogeneity of U.S. cities and the distribution of community members among areal units by considering their proportions above or below the city’s population [21]. Consequently, the measure can be precisely adapted to any urban area in the United States, as census tract data on racial/ethnic demographics is publicly available through the ACS five-year estimates.

### 8.2. Outcome Variable

In this study, the outcome variable was the mean life expectancy at birth for each of the four racialized/ethnic populations at the census tract level, established using life expectancy data from the CDC’s National Center for Health Statistics U.S. Small Area Life Expectancy Estimates Project (USALEEP), which was collected between 2010 and 2015. The outcome variable was examined by quartile, and the mean life expectancy of fourth quartile census tracts (i.e., tracts in which a group is most overrepresented) was compared to the state-level mean life expectancy for each group. The mean life expectancy at birth for Texas counties (*n* = 254) was established using data from County Health Rankings and Roadmaps, which were also collected from the USALEEP [38].

### 8.3. Control Variables

In the statistical models, three covariates were used to control for socioeconomic disparities at the census tract level. Socioeconomic characteristics were chosen as controls to implicate the possibility of factors beyond the socioeconomic gradient explaining life expectancy disparities. These variables were selected based on the Healthy People 2030 social determinants of health framework [39]. The covariates were employed to adjust for the extent to which the association between mean life expectancy and degree of segregation may be confounded by secondary factors related to social determinants of health. These factors include rates of (1) poverty, (2) uninsurance, and (3) high school completion. The data for these variables, for each census tract and each county in Texas, were obtained from the 2019 ACS 5-year estimates.

## 9. Data Analysis

The measure’s ability to predict the mean life expectancy at birth in Dallas census tracts was evaluated. However, due to the limitations of the data, it is important to note that although life expectancy estimates are disaggregated by racialized/ethnic community at the county level, they are aggregate estimates at the census tract level. Consequently, it is not possible to understand the association between segregation degree as calculated by the measure and life expectancy for each racialized/ethnic community. Instead, this analysis assessed the difference between the mean life expectancy at birth for census tracts with the highest degree of overrepresentation (fourth quartile) for a given community and the life expectancy at birth for that community at the state level (averaged across all Texas counties). The underlying assumption of this approach was that this difference would indicate the association between segregation degree and life expectancy. If census tracts with the highest relative overrepresentation of a given racialized/ethnic community had lower life expectancy than those for that community at the state level, then segregation may play a significant role in that difference. While the use of aggregated data at the census tract level is a clear limitation of this analysis, the decision was made to use these variables to demonstrate the possibility of disparities at the neighborhood level and expose the need for health data disaggregated by racialized/ethnic group at a more granular level.

The measure was employed to determine the degree of segregation for communities (Hispanic, non-Hispanic white, non-Hispanic Black, and Asian) in Dallas census tracts (*n* = 303). Tracts with the highest degrees of relative segregation were identified by examining each community by quartiles. The first quartile consisted of census tracts where a community was relatively underrepresented, while the fourth quartile comprised tracts where a community was relatively overrepresented. As this study focused on measuring and understanding the effects of structurally influenced concentration of racialized/ethnic communities at the neighborhood level, the analysis centered on the fourth quartile census tracts. However, results for all quartiles are provided in Table 3, Table 4, Table 5 and Table 6.

A descriptive analysis was conducted to estimate the mean degree of segregation, life expectancy, and control variables for each community’s census tracts by quartile and for Texas counties (Table 3, Table 4, Table 5 and Table 6). Multivariable linear regression was used to assess the significance of the difference between the fourth quartile census tracts’ life expectancy for each community and the state mean life expectancy for that community, controlling for rates of poverty, uninsurance, and high school completion (See Table 7). Four separate models were constructed to assess the relationship between the predictor and outcome variables for each racialized/ethnic community. Poverty, uninsurance, and high school completion rates were controlled for in all models. Listwise deletion was used to exclude Texas counties that did not report life expectancy for each racialized/ethnic community from the regression samples. This method aligns with the established procedures outlined by the County Health Rankings and Roadmaps [38], ensuring consistency with recognized public health research standards. All analyses were conducted in Stata 17.

## 10. Results

The unadjusted means of segregation degree and life expectancy at birth for census tracts by quartile and Texas counties, stratified by racialized/ethnic community, are presented in Table 3, Table 4, Table 5 and Table 6. The means for segregation degree of fourth quartile tracts (those with the most overrepresentation for each group) were 32.87 for the Hispanic community (SE = 1.33), 47.84 for the white community (SE = 1.09), 33.79 for the Black community (SE = 1.82), and 8.77 for the Asian community (SE = 1.11). Unadjusted means for life expectancy at birth for each community at the state level were 80.82 years (SE = 0.16) for those who identified as Hispanic, 76.25 years (SE = 0.22) for those who identified as white, 73.62 years (SE = 0.37) for those who identified as Black, and 87.99 years (SE = 0.77) for those who identified as Asian. Unadjusted means for life expectancy at birth for fourth quartile tracts by racialized group were 76.67 years (SE = 0.27) for those who identified as Hispanic, 81.15 years (SE = 0.25) for those who identified as white, 73.38 years (SE = 0.48) for those who identified as Black, and 79.99 years (SE = 0.34) for those who identified as Asian.

As relative overrepresentation of the Hispanic community increased from the first to fourth quartiles, unadjusted mean life expectancy decreased by 3.25 years, from 79.92 years (SE = 0.54) in the first quartile to 76.67 years in the fourth quartile (SE = 0.27). The unadjusted life expectancy decrease was more dramatic (7.02 years) with increased Black segregation degree, from 80.40 years (SE = 0.31) in the first quartile to 73.38 years (SE = 0.48) in the fourth quartile tracts. Unadjusted means for life expectancy increased with increased white segregation degree by 6.89 years, from 74.26 years (SE = 0.44) in the first quartile to 81.15 years (SE = 0.25) in the fourth quartile, and for Asian segregation degree by 5.74 years, from 74.25 years (SE = 0.45) in the first quartile to 79.99 years (SE = 0.34) in the fourth quartile.

Unadjusted means of covariates for Texas and census tracts by quartile, stratified by racialized/ethnic community, are also provided in Table 3, Table 4, Table 5 and Table 6. Comparing unadjusted means between first and fourth quartile tracts demonstrated increases in poverty and uninsurance as Hispanic segregation degree increased. Poverty rose by 13.6 percentage points (*pp*), while uninsurance rose by 25.0 *pp*. Uninsurance rates were highest in fourth quartile tracts relatively overrepresented by the Hispanic community (M = 33.1, SE = 0.97), and high school diploma rates were lowest (M = 56.7, SE = 1.16). Poverty rates for fourth quartile tracts were 7.2 *pp* higher than the rate in Texas. Similarly, fourth quartile tracts were 11.3 *pp* higher than the state uninsurance mean. Conversely, high school diploma rates were 25.1 *pp* lower than the state mean.

Differences in unadjusted means were pronounced between first and fourth quartiles for the white community. Poverty decreased by 23.0 *pp* and was found to be 9.8 *pp* lower than the state mean. A similar pattern was observed for uninsurance rates, which fell by 21.9 *pp* between first and fourth quartiles and were lower than the state rate by 15.2 *pp*. Conversely, the high school diploma rate increased by 31.8 *pp* as the white community increased between quartiles, which was the highest among the four communities (M = 96.49, SE = 0.54) and was 14.7 *pp* higher than the state rate.

Poverty and uninsurance rates increased with relative Black overrepresentation. Poverty rose by 18.4 *pp* between the first and fourth quartiles and was 12.7 *pp* above the state poverty rate. Fourth quartile tracts had the highest poverty rate among communities (M = 28.0%, SE = 1.12) and was as high as 50.8%. Uninsurance also increased with Black representation in a census tract (8.2 *pp*) and was 3.1 *pp* higher than the state rate. Conversely, high school diploma rates decreased by 4.6 *pp* as relative Black representation increased and was 5.4 *pp* below the state mean.

Increased Asian representation in tracts followed patterns of decreased poverty and uninsurance and increased high school diploma rates. Poverty decreased by 14.2 *pp* as Asian representation increased in census tracts and was 3.0 *pp* above the state rate. Uninsurance decreased by 13.7 *pp*, with rates 7.0 *pp* below the Texas mean. High school diploma rates increased by 24.8 *pp* between first and fourth quartile tracts and were 8.7 *pp* higher than the state rate.

Results for linear models are presented in Table 7. When adjusted for rates of poverty, uninsurance, and high school diploma, life expectancy for the white community at the state level was lower than mean life expectancy in tracts where the white community is relatively overrepresented (*b* = −3.73; 95% CI [−5.20, −2.27], *p* < 0.001). Adjusted models for life expectancy for the Black community also demonstrated that the community’s life expectancy in Texas was lower than the mean life expectancy in fourth quartile tracts where the Black community is overrepresented (*b* = −2.02; 95% CI [−3.60, −0.44]; *p* = 0.01). However, life expectancy for the Asian community in Texas was higher than the mean life expectancy for tracts where the Asian community is relatively overrepresented (*b* = 8.92, 95% CI [6.96, 10.07], *p* < 0.001). Findings were not significant for the Hispanic community.

## 11. Discussion

The objectives of this study were twofold: first, to develop and test a novel measure that estimates the degree of segregation at the census tract level, and second, to assess for significant disparities in life expectancy between relative overrepresentation and state-level estimates for the corresponding racialized/ethnic group. We predicted that greater overrepresentation of a specific racialized/ethnic community within a census tract, relative to their proportion of the city population, would correlate with significant variations in the tract’s mean life expectancy compared to the community’s life expectancy at the state level. Furthermore, we anticipated that the direction and magnitude of these differences would differ across groups.

Linear models did not yield significant results for the Hispanic population, implying that census tract residence may not account for the 4.2-year gap in life expectancy between fourth quartile Dallas residents and the Hispanic population in Texas. However, the findings supported both hypotheses for the non-Hispanic Black, non-Hispanic white, and Asian communities. Unadjusted means suggested a decline in life expectancy as the relative proportion of the Black community increased within tracts. Yet, adjusted linear models revealed that the average life expectancy for the Black population in Texas was lower than in Dallas census tracts with a higher concentration of Black residents. In contrast, linear models indicated a significant decrease in life expectancy when comparing adjusted means of highly segregated tracts to the statewide life expectancy for the Asian community.

The variability in findings across racialized/ethnic groups suggests that overrepresentation in a census tract may have both beneficial and adverse effects on health outcomes. For example, the observation that life expectancy is higher in census tracts with a high concentration of Black residents compared to the state average for the Black population supports the understanding of the health-protective advantages of increased social capital, community cohesion, and robust community ties [9,13,29]. As illustrated in Table 5, this finding is particularly noteworthy considering that socioeconomic indicators decline as the proportion of Black residents increases in a tract (e.g., increased poverty and uninsurance rates and decreased high school completion rates). This seeming paradox underscores the complex interplay between place, social determinants, and health outcomes, suggesting that the lived experience of residing in a racially or ethnically concentrated tract may buffer against some of the negative impacts of structural inequities. Further research is needed to elucidate the specific mechanisms through which racial/ethnic density influences health across different groups, with the ultimate goal of promoting health equity.

In contrast, the finding that life expectancy is lower in census tracts with high Asian ethnic density, despite the association of Asian overrepresentation with lower poverty and uninsurance rates and higher educational attainment (Table 6), suggests that factors beyond socioeconomic status may be influencing health outcomes in these communities. One potential explanation for this counterintuitive finding is the presence of densely populated census tracts where agencies resettled individuals entering the United States with refugee status, many of whom come from Asian countries such as Burma. The migration experience of forced displacement, which often involves trauma and acculturation stress, may act as a powerful social determinant of health that transcends the protective effects typically associated with higher socioeconomic status [40]. This underscores the importance of examining the intersections between race/ethnicity, immigration status, and other social identities when investigating the health impact of ethnic density. It also highlights the need for more granular data that captures the diversity within broad racialized/ethnic categories, as the experiences and health outcomes of recent refugees likely differ from Asian American communities. Future research should aim to disentangle the effects of ethnic density by factors such as nativity status, length of residency in the United States, and specific country of origin to develop a more nuanced understanding of how these variables interact with place-based characteristics to shape health disparities.

When interpreting the contrasting relationship between census tract composition and life expectancy for Black and Asian communities, it is essential to consider individuals’ complex connections to place. For example, Texas has the highest proportion of rural residents in the United States, with 22.8% of its 254 counties being exclusively rural [41]. This context raises important questions about the influence of rurality on the observed disparity in life expectancy between Dallas census tracts with a high proportion of Dallas residents and the overall life expectancy for Texans who identify as Black. The present study’s findings, which controlled for socioeconomic factors, indicate that residing in certain urban census tracts may be associated with longer life expectancy. These results emphasize the need to consider recent longitudinal research demonstrating significant associations between experiences of racial discrimination and accelerated health decline among Black individuals in the United States Americans over time [42].

Texans who identify as Black make up eight percent of the rural population and 13 percent of the urban population [43]. Considering that this study found higher life expectancy among Black residents of urban census tracts with greater Black representation compared to the statewide average, it is imperative to include the experiences of people of color living in rural and unincorporated areas in health disparities research. Examining the potential differences in exposure to anti-Black discrimination between rural and urban environments, as well as incorporating structural factors such as access to health-related resources as covariates, represent critical directions for future research to better understand these disparities.

The results also revealed an association between higher proportions of white residents in a census tract and longer life expectancy. Given that patterns of social vulnerability often align with historical redlining practices [5], this finding highlights the crucial issue of place as a locus of power (i.e., racialized privilege), illustrating that place is not a neutral backdrop. Rather, in these tracts, “whiteness functions as a spatialized and structured advantage” [44], jointly shaping place and race within an urban context [7]. Moreover, these findings underscore the “place-bound character of white identity in the United States” [44] and how historically racialized divides maintain color lines that shape various facets of urban life, benefiting some while disadvantaging others.

While this study primarily focuses on the influence of place as a determinant of health outcomes [45], it is important to acknowledge the broader scientific consensus on the interplay between biological and environmental factors. This research aligns with contemporary views in public health that recognize health as the result of a complex interplay between genetic predispositions and a range of social, economic, and environmental factors [46]. Although our study does not directly measure biological variables, the significant associations found between life expectancy and place-based factors highlight the potential interactions that these environmental influences might have with biological aspects. Further research could explore these interactions in more detail, providing a more holistic understanding of how individual and collective health outcomes are shaped. This perspective encourages a multidisciplinary approach, integrating insights from genetics, sociology, and environmental science to enrich our understanding of health disparities.

## 12. Implications

The primary deliverable of this study, a measure of segregation at the census tract level, addresses the shortcomings of standard segregation indices which measure segregation across large geographic areas and mask the observation of key place-based mechanisms implicated as drivers of health disparities. By calculating segregation degree at a more granular level and examining its association with life expectancy, this study exposes the possibility of factors beyond socioeconomic characteristics influencing health outcomes in relatively segregated neighborhoods.

The key implications of these findings point to several potential strategies for future research. First, the segregation measure piloted here can be precisely adapted to examine health disparities in any urban area in the United States, as the census tract demographic data used in its calculation is publicly available. Testing the measure’s ability to isolate place-based disparities by applying it to a larger sample of highly segregated census tracts across multiple cities is a critical next step. Second, a clear limitation exposed by this study is the lack of publicly available health outcome data disaggregated by race/ethnicity at the census tract level. Addressing this data inequity is essential for accurately measuring the relationship between neighborhood segregation and health for specific racialized/ethnic communities. Obtaining and analyzing disaggregated data, potentially through healthcare claims, hospitalization records or original data collection, is needed.

Furthermore, while this study controlled for socioeconomic factors, future research should examine other place-based covariates that may influence the segregation–health relationship, such as access to healthcare, healthy food, green space, and exposure to pollutants. Disaggregating these variables by racialized/ethnic group at the census tract level would provide a clearer understanding of living conditions for each community. Additionally, the quantitative approach used here should be complemented by qualitative research that captures the lived experience of residing in segregated neighborhoods. Methods such as interviews, focus groups, photovoice, and neighborhood mapping can offer an explanatory dimension that elucidates the mechanisms through which place influences health, as well as differences in the experiences of segregation between communities.

Future research should also consider the impact of the COVID-19 pandemic on the relationship between segregation and health disparities. The pandemic has disproportionately affected communities of color, particularly those living in segregated neighborhoods, leading to widening disparities in life expectancies [47,48]. Analyzing post-pandemic data will be important for understanding how the health consequences of segregation may have been exacerbated by the crisis and for informing targeted interventions and policies to address these disparities. Considering recent findings by Fraser et al., which underscore the significant role of social capital in mitigating COVID-19 outcomes at local levels, future research should also consider the influence of neighborhood networks and support systems on health outcomes, exploring how enhanced community cohesion and robust social infrastructures can mitigate health disparities in segregated neighborhoods [49].

Finally, capitalizing on the stability of census tract boundaries over time, this measure can be used to conduct longitudinal studies examining changes in relative overrepresentation and health outcomes. This approach is critical for evaluating the impact of policies and interventions aimed at promoting integration and health equity. The representation measure presented here provides a more place-centered approach to examining health disparities. The strategies outlined above would build on this work, leveraging more granular, disaggregated data and novel methodologies to deepen our understanding of the pathways connecting segregation and health. Ultimately, findings from such research can directly inform place-based interventions and policies that target the structural roots of racialized health inequities.

## 13. Limitations

Several limitations should be considered when interpreting these results. Initially, the accuracy of the model was affected by the absence of data from certain counties. While this method facilitates the use of complete cases, it may also impact the generalizability of the findings. Future studies could benefit from exploring alternative imputation techniques to handle missing data, potentially providing a more comprehensive view of the impact across all counties. Additionally, conducting the study within a single geographic area, Dallas, Texas, may limit the generalizability of the findings to other regions, warranting caution when applying these results more broadly.

Additionally, privacy regulations concerning personal health records prevent the disaggregation of the study’s outcome variable by racialized/ethnic groups at the census tract level. Consequently, this study could not assess the life expectancy variations among different groups within individual census tracts. Rather, the study assessed life expectancy using aggregated data, represented by the average life expectancy in areas predominantly inhabited by certain groups. These data limitations, which pertain to issues of data justice, imply that until health data can be disaggregated and made publicly accessible, it will remain challenging to accurately determine how neighborhood-level segregation affects health outcomes.

Another noted limitation involves the reliance on the 2015–2019 American Community Survey (ACS) five-year estimates for constructing the index. These estimates are generally the most dependable for studying smaller populations. Yet, the accuracy of these estimates in reflecting the present population dynamics may be compromised due to the COVID-19 pandemic’s effects on both population and life expectancy metrics. However, the disturbances reported in the 2020 decennial census data collection [50] because of the pandemic reinforced the decision to utilize the 2015–2019 ACS data.

Another limitation pertains to potential biases in data selection. The ACS data, while comprehensive, does not fully account for the nuanced dynamics of migration and gentrification that may alter the demographic makeup of these areas over short periods. This limitation could influence the generalizability of our findings, particularly in highly dynamic urban settings. Additionally, the potential for selection bias exists due to the methodology used in data collection and the inherent limitations of the datasets employed. Such biases could skew interpretations, particularly in assessing the impact of segregation on health outcomes. Steps were taken to mitigate these biases, including rigorous statistical controls and the use of multivariable models that account for a range of socioeconomic variables. However, residual confounding might still be present, and findings should be interpreted with caution. Future studies should aim to include more granular data, ideally disaggregated by smaller demographic groups within census tracts, to better capture the nuances of health disparities influenced by segregation.

## 14. Conclusions

This study highlighted potential mechanisms beyond socioeconomic factors by evaluating segregation levels at the census tract level and exploring their association with life expectancy. By comparing mean life expectancy in census tracts where members of racialized/ethnic groups are relatively overrepresented to the state-level mean life expectancy for those groups, this research underscored the importance of considering place as a critical determinant of health. This approach draws on a theoretical framework rooted in critical place inquiry and builds upon the understanding that ZIP code is a more significant predictor of health outcomes than genetic code [45]. These findings suggest that the relationship between individuals and their environments plays a crucial role in understanding how segregation influences health outcomes differently across communities.

Future research should delve deeper into the underlying mechanisms of these associations. Specifically, studies could examine how environmental, social, and economic factors within segregated communities interact to influence health outcomes. Additionally, developing and evaluating community-based interventions aimed at mitigating health disparities in these settings could provide practical strategies for health improvement. By focusing on community-specific needs and resources, such interventions could address the unique challenges posed by residential segregation. This line of inquiry would help extend the current study’s findings and contribute to a more nuanced understanding of how place-based factors shape health disparities, ultimately informing targeted public health strategies.

## Figures and Tables

**Table 1 ijerph-21-00613-t001:** The 2020 values of formal segregation indices for Dallas, Texas.

Index		Value
Dissimilarity ^1^		
	White–Black/Black–white	64.5
	White–Hispanic/Hispanic–white	63.4
	White–Asian/Asian–white	35.5
	Black–Hispanic/Hispanic–Black	45.6
	Black–Asian/Asian–Black	59.4
	Hispanic–Asian/Asian–Hispanic	63.9
Isolation ^2^		
	White–white	54.2
	Black–Black	43.1
	Hispanic–Hispanic	59.4
	Asian–Asian	11.1
Exposure ^3^		
	Black–white	15.6
	Hispanic–white	16.1
	White–Hispanic	24.2
	Asian–white	41.4
	White–Asian	6.3
	White–Black	13.2
	Hispanic–Black	20.8
	Black–Hispanic	36.7
	Asian–Black	19.4
	Black–Asian	3.4
	Asian–Hispanic	26.2
	Hispanic–Asian	2.6

Note: Data source: Spatial Structures in the Social Sciences, Brown University. (2022). Diversity and disparities. American Communities Project. https://s4.ad.brown.edu/projects/diversity/segregation2020/Default.aspx (accessed on 5 May 2022). ^1^ Measures the proportion of a group that would have to move to a different census tract to achieve an even distribution of group members in the city. Values range from 0 to 100, with a value of 60 or greater indicating a high level of segregation. ^2^ Measures the percentage of same group population in a census tract where the average group member lives. Values range from 0 (group is dispersed) to 100 (group is entirely isolated). ^3^ Measures the possibility of contact between different group members. Values range from 0 to 100 with larger values indicating the typical group member lives in a census tract with a high percentage of same-group members.

**Table 2 ijerph-21-00613-t002:** Data sources.

Variable		Dataset	Data Source	Years	Geography
Predictor: Representation Measure					
	Census tract (A)	% Population of all people who were Hispanic or Latino	ACS ^1^	2015–2019	Census tract
		% Population of all people who were non-Hispanic white	ACS	2015–2019	Census tract
		% Population of all people who were non-Hispanic Black	ACS	2015–2019	Census tract
		% Population of all people who were Asian	ACS	2015–2019	Census tract
	City (B)	% Population of all people who were Hispanic or Latino	ACS	2015–2019	Census tract
		% Population of all people who were non-Hispanic white	ACS	2015–2019	Census tract
		% Population of all people who were non-Hispanic Black	ACS	2015–2019	Census tract
		% Population of all people who were Asian	ACS	2015–2019	Census tract
Outcome: Life Expectancy		Life expectancy at birth	USALEEP ^2^	2010–2015	Census tract and county
Covariates					
	Poverty	Estimated % of all people that are living in poverty	ACS	2016–2020	Census tract
	Uninsurance	Estimated % of all people without health insurance	ACS	2016–2020	Census tract
	High school diploma	Estimated % of people with at least a high school diploma	ACS	2016–2020	Census tract

Note: Segregation degree = A − B. ^1^ American Community Survey, U.S. Census Bureau. ^2^ U.S. Small-area Life Expectancy Estimates Project.

**Table 3 ijerph-21-00613-t003:** Unadjusted means of study variables for Hispanic community at quartile and state levels.

Variable		*n*	M	SE	95% CI	
					LL	UL
Segregation Degree						
	First Quartile	75	−33.06	0.42	−33.89	−32.23
	Second Quartile	77	−20.01	0.48	−20.95	−19.07
	Third Quartile	76	0.44	0.86	−1.26	2.14
	Fourth Quartile	75	32.87	1.33	30.26	35.48
Life Expectancy						
	Texas	170	80.82	0.43	79.98	81.67
	First Quartile	65	79.92	0.54	78.85	80.99
	Second Quartile	62	76.99	0.50	75.99	77.98
	Third Quartile	62	75.77	0.50	74.76	76.78
	Fourth Quartile	67	76.67	0.27	76.13	77.20
Poverty						
	Texas	254	15.32	0.40	14.54	16.11
	First Quartile	75	8.87	1.07	6.75	10.99
	Second Quartile	77	16.52	1.25	14.03	19.00
	Third Quartile	76	22.73	1.34	20.06	25.40
	Fourth Quartile	75	22.47	0.97	20.54	24.40
Uninsurance						
	Texas	254	21.79	0.26	21.27	22.30
	First Quartile	75	8.07	0.83	6.42	9.72
	Second Quartile	77	17.68	0.90	15.88	19.48
	Third Quartile	76	26.49	0.90	24.70	28.29
	Fourth Quartile	75	33.10	0.96	31.17	35.02
High School Diploma						
	Texas	254	81.79	0.53	80.74	82.84
	First Quartile	75	95.64	0.80	94.06	97.23
	Second Quartile	77	87.23	1.06	85.12	89.33
	Third Quartile	76	75.20	1.13	72.96	77.45
	Fourth Quartile	75	56.70	1.16	54.38	59.01

Note: First quartile tracts consist of residential census tracts where the Hispanic community is relatively underrepresented, when compared to their composition in the city’s total population. Fourth quartile tracts consist of those where the Hispanic community is relatively overrepresented.

**Table 4 ijerph-21-00613-t004:** Unadjusted means of study variables for non-Hispanic white community at quartile and state levels.

Variable		*n*	M	SE	95% CI	
					LL	UL
Segregation Degree						
	First Quartile	76	−25.11	0.25	−25.61	−24.61
	Second Quartile	76	−13.59	0.60	−14.76	−12.42
	Third Quartile	76	12.72	1.12	10.52	14.93
	Fourth Quartile	75	47.84	1.09	45.68	49.99
Life Expectancy						
	Texas	180	76.25	0.22	75.81	76.68
	First Quartile	73	74.26	0.44	73.38	75.14
	Second Quartile	58	75.75	0.43	74.90	76.61
	Third Quartile	58	78.46	0.28	77.90	79.02
	Fourth Quartile	67	81.15	0.25	80.65	81.65
Poverty						
	Texas	254	15.32	0.40	14.54	16.11
	First Quartile	76	28.57	1.06	26.45	30.69
	Second Quartile	76	22.66	1.00	20.67	24.66
	Third Quartile	76	13.67	0.82	12.04	15.31
	Fourth Quartile	75	5.55	0.48	4.60	6.51
Uninsurance						
	Texas	254	21.79	0.26	21.27	22.30
	First Quartile	76	28.51	0.88	26.75	30.27
	Second Quartile	76	30.30	0.96	28.38	32.22
	Third Quartile	76	19.67	1.11	17.46	21.88
	Fourth Quartile	75	6.64	0.58	5.48	7.80
High School Diploma						
	Texas	254	81.79	0.53	80.74	82.84
	First Quartile	76	64.73	1.37	61.99	67.46
	Second Quartile	76	69.47	1.68	66.12	72.83
	Third Quartile	76	84.49	1.51	81.48	87.51
	Fourth Quartile	75	96.49	0.53	95.43	97.56

Note: First quartile tracts consist of residential census tracts where the non-Hispanic white community is relatively underrepresented, when compared to their composition in the city’s total population. Fourth quartile tracts consist of those where the non-Hispanic white community is relatively overrepresented.

**Table 5 ijerph-21-00613-t005:** Unadjusted means of study variables for non-Hispanic Black community at quartile and state levels.

Variable		*n*	M	SE	95% CI	
					LL	UL
Segregation Degree						
	First Quartile	76	−22.19	0.15	−22.48	−21.89
	Second Quartile	76	−15.41	0.34	−16.08	−14.75
	Third Quartile	76	−0.36	0.56	−1.47	0.75
	Fourth Quartile	75	33.79	1.82	30.21	37.37
Life Expectancy						
	Texas	99	73.62	0.37	72.89	74.36
	First Quartile	68	80.40	0.31	79.78	81.01
	Second Quartile	67	78.61	0.32	77.98	79.24
	Third Quartile	55	76.83	0.37	76.10	77.56
	Fourth Quartile	66	73.38	0.48	72.43	74.33
Poverty						
	Texas	254	15.32	0.40	14.54	16.11
	First Quartile	76	9.61	1.03	7.55	11.66
	Second Quartile	76	14.74	1.17	12.40	17.08
	Third Quartile	76	18.41	1.02	16.38	20.44
	Fourth Quartile	75	28.00	1.12	25.78	30.23
Uninsurance						
	Texas	254	21.79	0.26	21.27	22.30
	First Quartile	76	16.63	1.89	12.86	20.40
	Second Quartile	76	18.95	1.34	16.29	21.61
	Third Quartile	76	24.91	1.11	22.69	27.12
	Fourth Quartile	75	24.87	0.77	23.33	26.41
High School Diploma						
	Texas	254	81.79	0.53	80.74	82.84
	First Quartile	76	80.96	2.55	75.87	86.05
	Second Quartile	76	80.39	2.10	76.20	84.58
	Third Quartile	76	77.21	1.77	73.68	80.73
	Fourth Quartile	75	76.36	1.18	74.00	78.71

Note: First quartile tracts consist of residential census tracts where the non-Hispanic Black community is relatively underrepresented, when compared to their composition in the city’s total population. Fourth quartile tracts consist of those where the non-Hispanic Black community is relatively overrepresented.

**Table 6 ijerph-21-00613-t006:** Unadjusted means of study variables for Asian community at quartile and state levels.

Variable		*n*	M	SE	95% CI	
					LL	UL
Segregation Degree						
	First Quartile	75	−3.39	0.00	−3.39	−3.39
	Second Quartile	78	−2.48	0.06	−2.60	−2.36
	Third Quartile	75	0.05	0.10	−0.16	0.26
	Fourth Quartile	75	8.77	1.11	6.59	10.95
Life Expectancy						
	Texas	38	87.99	0.77	86.44	89.55
	First Quartile	69	74.25	0.44	73.36	75.14
	Second Quartile	70	76.99	0.46	76.07	77.91
	Third Quartile	63	78.90	0.38	78.14	79.65
	Fourth Quartile	54	79.99	0.34	79.31	80.66
Poverty						
	Texas	254	15.32	0.40	14.54	16.11
	First Quartile	75	26.59	1.15	24.31	28.88
	Second Quartile	78	17.46	1.11	15.24	19.68
	Third Quartile	75	14.23	1.31	11.62	16.84
	Fourth Quartile	75	12.35	1.14	10.09	14.62
Uninsurance						
	Texas	254	21.79	0.26	21.27	22.30
	First Quartile	75	28.50	0.99	26.52	30.49
	Second Quartile	78	24.36	1.40	21.57	27.15
	Third Quartile	75	17.54	1.33	14.89	20.20
	Fourth Quartile	75	14.78	1.28	12.24	17.32
High School Diploma						
	Texas	254	81.79	0.53	80.74	82.84
	First Quartile	75	65.64	1.51	62.62	68.66
	Second Quartile	78	72.71	1.99	68.74	76.68
	Third Quartile	75	86.36	1.50	83.37	89.34
	Fourth Quartile	75	90.48	1.32	87.85	93.12

Note: First quartile tracts consist of residential census tracts where the non-Hispanic Black community is relatively underrepresented, when compared to their composition in the city’s total population. Fourth quartile tracts consist of those where the non-Hispanic Black community is relatively overrepresented.

**Table 7 ijerph-21-00613-t007:** Adjusted estimates of differences between mean life expectancy at birth for fourth quartile census tracts and Texas, stratified by race/ethnicity (*n* = 300).

Life Expectancy		*b*	SE	*t*	*p*	95% CI	
						LL	UL
Hispanic		0.49	1.21	0.41	0.686	−1.89	2.87
	Poverty	−0.06	0.06	−1.15	0.251	−0.17	0.05
	Uninsurance	0.10	0.08	1.24	0.216	−0.06	0.26
	High school diploma	0.17	0.05	3.23	0.001	0.07	0.28
Non-Hispanic white		−3.73	0.74	−5.02	**<0.001**	−50.20	−2.27
	Poverty	−0.02	0.05	−0.36	0.717	−0.11	0.07
	Uninsurance	−0.09	0.06	−1.70	0.091	−0.20	0.02
	High school diploma	−0.03	0.04	−0.73	0.468	−0.11	0.05
Non-Hispanic Black		−2.02	0.80	−2.53	**0.012**	−30.60	−0.44
	Poverty	−0.14	0.05	−2.78	0.006	−0.24	−0.04
	Uninsurance	0.09	0.08	1.15	0.250	−0.07	0.25
	High school diploma	0.08	0.06	1.34	0.182	−0.04	0.20
Asian		8.52	0.78	10.90	**<0.001**	6.96	10.07
	Poverty	0.01	0.07	0.12	0.905	−0.12	0.14
	Uninsurance	−0.20	0.09	−2.27	0.026	−0.37	−0.02
	High school diploma	−0.11	0.08	−1.36	0.179	−0.28	0.05

Note: models adjusted for rates of poverty, uninsurance, and high school diploma. Bold indicates statistical significance at a *p* < 0.05 threshold.

## Data Availability

Data are contained within the article.

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
