# Peer review of "Piloting a Measure of Segregation at the Census Tract Level: Associations with Place and Racial/Ethnic Disparities in Life Expectancy"

_ijerph, 2024, doi:10.3390/ijerph21050613_

Round 1
Reviewer 1 Report
Comments and Suggestions for Authors
The article deals with a very interesting and necessary subject. It is very well written and clear. The summary is comprehensive and explanatory, which is very helpful to the reader.
The sections into which the paper is divided help to understand the research and to follow it in detail. In this sense, I found section 12, implications, very gratifying, where an attempt is made to go beyond the results obtained.
The results obtained are clear and well explained. I believe that the software used to carry out the statistical analysis has not been indicated, it would be convenient to put it.
The discussion, although I find it a little brief, meets the minimum requirements. It would be appreciated if the authors could expand it a little more to deepen the results.
I would like to suggest to the authors the introduction of an outline of the implications of the analysis performed. This outline could help those interested to see clearly the possible strategies to be developed. I mean to indicate the main variables found by the authors.
Finally, I would like to suggest to the authors that they make the effort to rework the paper, especially pages 3-11 and 17-25. The article has copied, literally, Kitchens' thesis and I think it would be good to change this.
Thank you so much!
Reviewer 2 Report
Comments and Suggestions for Authors
Dear authors, I have carefully reviewed your article and would like to express my appreciation for your significant contribution to understanding health disparities based on place and residential segregation. Your study highlights the importance of considering not only socioeconomic factors but also place-related mechanisms in determining the health of racial/ethnic populations. I especially value your theoretical approach based on critical place research, which emphasizes the need to understand how the physical and social environment influences health disparities.
The most important issue that needs to be addressed is the lack of updated information on the current scientific developments related to the topic of the manuscript. The work is methodologically sound, clear and robust. However, the findings need to be explained in the context of current scientific knowledge, especially after the changes brought about by the pandemic.
Some points need to be addressed:
1. Limitations and scope of the findings: The discussion of study limitations and the scope of the findings could be expanded to address aspects such as potential biases in data selection and how they might affect the interpretation of results.
2. Ethical considerations: It would be beneficial to include an additional section addressing ethical considerations related to data use and the protection of participant privacy, especially regarding data aggregation at the census tract level.
3. Recommendations for future research: The conclusions section could be enriched by providing specific recommendations for future research in this field, such as exploring underlying mechanisms of the associations found and developing community-based interventions to address health disparities.
4. Considering that it is mentioned that data will be removed when it is nonexistent, it is convenient to indicate how many data points were removed and how this may affect representativeness in measurements. Similarly, since there is currently acceptance in the scientific community of the interaction between internal (biological) and external (social, environmental) factors, it is worth discussing the idea that postal code weighs more than biological code, because health exists on a body and its interaction with external factors. In this sense, I suggest placing the discussion on interaction and its understanding, rather than on what weighs more within the health outcome. Likewise, review reference 46 cited to support the idea, as it is not referenced. I also suggest updating, in the introduction, how the 1988 reference theory remains relevant or is still used in other territories and what the findings have been. Following the two questions addressed in your study, in the introduction you can specify how they are found in other parts of the world in relation to them. This is to situate your research in the current scientific context.
5. I also wondered if it was necessary to present the comparison in Table 1 with the rest of the United States to understand how Dallas compares to the country.
6. In the discussion, it is highly recommended to discuss with recent studies, preferably from the past year, as I find the list of references out of context, especially due to health outcomes after the pandemic. I suggest reviewing articles in The Lancet or the most updated statistics on the post-2022 health dynamics. Overall, your study presents a valuable contribution to the field of health disparities and research on residential segregation. We look forward to reviewing a revised version of the manuscript that addresses these points.
Reviewer 3 Report
Comments and Suggestions for Authors
The study focuses on racial/ethnic segregation, but other social determinants of health, such as income inequality, access to healthcare, and educational opportunities, also play crucial roles in health disparities. An analysis that includes these factors could provide a more comprehensive understanding of the complex interplay between segregation and health outcome .This suggestion is for future scientific studies
Reviewer 4 Report
Comments and Suggestions for Authors
First, thank you for the opportunity to review the article “Pilot of a Census-Level Segregation Measure: Associations with Place and Racial/Ethnic Disparities in Life Expectancy.” First of all, thank you for your work for the relevance and importance of the topic and for the stated objective “to expose the need to consider the possibility of neighborhood mechanisms beyond socioeconomic characteristics as an important determinant of health and to draw attention to the importance to critically engage placeholders in examinations of racial and ethnic health disparities.” Both the title and the summary allow us to achieve a global vision of the study and invite it to be read, although a structure of the summary that includes a small section regarding the methodology used would be appreciated.
The introduction builds a theoretical framework regarding residential segregation and its relationship with health inequality, as well as the importance of conducting measurements at the neighborhood level. Different previous bibliography regarding the topic is listed. Can the importance of neighborhood networks or support at different times be included as a supporting factor? This aspect is then valued in the discussion.
Methodology. The study is justified in a single geographical area. Should this be considered a limitation of the study? It should be considered explaining in this section the methodological aspects: data collection, selection of variables (poverty rate, insurance, etc.). For this reason, tables 3 to 6 should be included as annexes and indicate the most relevant aspects of the results.
I believe that the results should be structured following the aspects of the methodology. It would be convenient to rewrite some paragraphs so that they allow value to be given to the relationship with different variables such as level of education, etc. On the other hand, has the importance of gender in the difference in the level of health or life expectancy been assessed?
The discussion is clear and responds to the objective of the study. However, a reference to different bibliographic articles that could compare similar initiatives would be appreciated.
Regarding the conclusions, it includes bibliographic references that I consider should be included in the discussion of the results. Also indicate that reference [46] does not appear in the bibliographical references.
References (44) include different reports. It would be appreciated, as previously mentioned, to include some current reference regarding “residential segregation” and life expectancy, level of health, etc.
Round 2
Reviewer 2 Report
Comments and Suggestions for Authors
Dear authors I find the manuscript much more scientifically accurate, thank you very much for taking the time to improve it.